# A 14-Day Plant-Based Dietary Intervention Modulates the Plasma Levels of Rheumatoid Arthritis-Associated MicroRNAs: A Bioinformatics-Guided Pilot Study

**DOI:** 10.3390/nu17132222

**Published:** 2025-07-04

**Authors:** Mario Peña-Peña, Elyzabeth Bermúdez-Benítez, José L. Sánchez-Gloria, Karla M. Rada, Mauricio Mora-Ramírez, Luis M. Amezcua-Guerra, Martha A. Ballinas-Verdugo, Claudia Tavera-Alonso, Carlos A. Guzmán-Martín, Leonor Jacobo-Albavera, Aarón Domínguez-López, Rogelio F. Jiménez-Ortega, Luis H. Silveira, Laura A. Martínez-Martínez, Fausto Sánchez-Muñoz

**Affiliations:** 1Sección de Estudios de Posgrado, Escuela Superior de Medicina, Instituto Politécnico Nacional, Mexico City 11340, Mexico; marionutricion2017@gmail.com (M.P.-P.); michelrp36@gmail.com (K.M.R.); aadominguezl@yahoo.com.mx (A.D.-L.); 2Departamento de Fisiología, Instituto Nacional de Cardiología Ignacio Chávez, Mexico City 14080, Mexico; gmcarlos93@gmail.com; 3Departamento de Reumatología, Instituto Nacional de Cardiología Ignacio Chávez, Mexico City 14080, Mexico; elymars@hotmail.com (E.B.-B.); luis_hsil@yahoo.com (L.H.S.); 4Department of Internal Medicine, Division of Nephrology, Rush University Medical Center, Chicago, IL 60612, USA; jose_sanchez@rush.edu; 5Departamento de Medicina Interna, Torre Médica (Grupo Dalinde), Centro Médico Dalinde, Tuxpan No. 29, Roma Sur, Cuauhtémoc, Mexico City 06760, Mexico; maucartney08@hotmail.com; 6Departamento de Atención a la Salud, Universidad Autónoma Metropolitana Xochimilco, Mexico City 04960, Mexico; lmamezcuag@gmail.com; 7Departamento de Inmunología, Instituto Nacional de Cardiología Ignacio Chávez, Mexico City 14080, Mexico; ballinasv75@gmail.com; 8Departamento de Bioquímica, Instituto Nacional de Cardiología Ignacio Chávez, Mexico City 14080, Mexico; 9Laboratorio Central, Instituto Nacional de Cardiología Ignacio Chávez, Mexico City 14080, Mexico; taveramuc@yahoo.com.mx; 10Doctorado en Ciencias Biológicas y de la Salud, Universidad Autónoma Metropolitana, Mexico City 04960, Mexico; 11Laboratorio de Genómica de Enfermedades Cardiovasculares, Instituto Nacional de Medicina Genómica, Mexico City 14610, Mexico; ljacobo@inmegen.gob.mx; 12Departamento de Medicina Genómica, Instituto Nacional de Rehabilitación (INR), Mexico City 14389, Mexico; rogeliofrank.jimenez@uneve.edu.mx

**Keywords:** rheumatoid arthritis, plant-based diet, microRNAs, bioinformatics, DAS28, inflammation, nutrition intervention, miR-26a-5p, miR-125a-5p, miR-155-5p

## Abstract

**Background/Objectives**: MicroRNAs (miRNAs) have emerged as molecular mediators involved in the pathogenesis of rheumatoid arthritis (RA). Given the influence of diet on gene expression and inflammation, plant-based diets represent a potential non-pharmacological strategy for modulating disease activity. This study aimed to explore and validate, through a bioinformatic-guided pilot approach, the regulation of miRNAs associated with RA in response to a 14-day plant-based dietary intervention. **Methods**: Candidate miRNAs were identified through differential expression analysis of the GSE124373 dataset using GEO2R and were further supported by a literature review. Target gene prediction and functional enrichment analyses were conducted to assess the biological relevance of these findings. Twenty-three RA patients followed a plant-based diet for 14 days. The clinical activity (DAS28-CRP), biochemical markers, and plasma expression of five selected miRNAs (miR-26a-5p, miR-125a-5p, miR-125b-5p, miR-146a-5p, and miR-155-5p) were evaluated before and after the intervention using RT-qPCR. **Results**: Significant reductions were observed in DAS28-CRP scores, C-reactive protein, glucose, and lipid levels after 14 days of intervention. Three of the five miRNAs (miR-26a-5p, miR-125a-5p, and miR-155-5p) were significantly downregulated post-intervention. Bioinformatic analyses indicated that these miRNAs regulate immune–inflammatory pathways relevant to RA pathogenesis. **Conclusions**: This pilot study suggests that a short-term plant-based dietary intervention may modulate circulating miRNAs and improve clinical and biochemical parameters in RA patients. These findings support further research into dietary strategies as complementary approaches for RA management.

## 1. Introduction

Rheumatoid arthritis is a chronic, systemic autoimmune disease characterized by symmetrical polyarthritis, morning stiffness, and joint swelling, commonly involving the small joints of the hands and feet [1]. It affects approximately 0.46% of the global population, with a higher prevalence in women (3:1), and represents a significant global health burden [2]. While conventional pharmacologic treatments remain the cornerstone of RA management [3], emerging evidence suggests that dietary interventions may offer beneficial adjunctive effects [4,5,6]. Poor-quality Western dietary patterns characterized by the high intake of processed foods, refined sugars, and saturated fats have been associated with increased inflammation and worsening of RA symptoms. In contrast, diets rich in anti-inflammatory components appear to have protective effects [7,8,9]. Among these, plant-based dietary patterns (PBDs) such as vegetarian and vegan diets have gained attention as potential therapeutic strategies for patients with RA [9,10]. PBDs emphasize the consumption of vegetables, fruits, legumes, and whole grains while limiting animal products, processed foods, and added sugars, resulting in a distinct and nutrient-dense profile [11,12].

In this regard, PBDs have shown promising effects in several inflammatory and autoimmune conditions, including RA [13]. Short-term studies indicate that these diets may reduce the circulating levels of pro-inflammatory proteins and eicosanoids, key mediators in RA pathogenesis [14]. By enhancing the intake of anti-inflammatory foods, PBDs may downregulate cytokine and chemokine expression, offering symptom relief at a molecular level [15]. Additionally, diets that minimize processed carbohydrates and saturated fats while incorporating healthy fats such as those found in olive oil and fish can positively influence inflammation-related signaling pathways in RA [16]. Despite these encouraging findings, the specific molecular mechanisms by which PBDs influence RA-associated inflammation remain incompletely understood.

In this sense, microRNAs (miRNAs) have emerged as key regulators in several autoimmune and inflammatory scenarios [17]. miRNAs are small non-coding RNAs approximately 22 nucleotides in length that post-transcriptionally regulate gene expression, and interestingly, they have recently been associated with dietary patterns, including anti-inflammatory diets and PBDs [18]. In addition, over the past decades, miRNAs have emerged as crucial molecular regulators in the pathogenesis of RA, influencing immune activation, joint destruction, and even therapeutic response [19]. Among the most studied miRNAs in RA are miR-16, miR-26a, miR-125a/b, miR-146a, and miR-155, which have demonstrated diagnostic, prognostic, and mechanistic relevance due to their regulatory effects on inflammatory signaling pathways [19,20]. Despite growing interest in the intersection of nutrition and molecular regulation, it remains unclear whether PBDs can modulate the circulating levels of RA-associated miRNAs.

Therefore, this pilot study aimed to explore and validate, through a bioinformatics-guided approach, the regulation of key RA-related miRNAs in response to a 14-day PBD intervention and to evaluate their potential relationship with changes in clinical and biochemical parameters in patients with rheumatoid arthritis.

## 2. Materials and Methods

### 2.1. Patient Recruitment

The present study was approved by the research ethics committee of the Instituto Nacional de Cardiología Ignacio Chávez, protocol registration INCAR-DG-DI-CI-DICT-042-2021. Additionally, it complied with international laws, Good Medical Practice Guidelines, and the Declaration of Helsinki. This study was also registered at ClinicalTrial.gov (Registered 29 August 2023, identifier: NCT05911880; www.clinicaltrials.gov).

Twenty-three patients (aged ≥18 years) diagnosed with RA according to the 2010 criteria of the American College of Rheumatology/European Alliance of Associations for Rheumatology (ACR/EULAR) [21] were recruited consecutively between December 2021 and June 2023 at the Rheumatology outpatient clinic. Disease activity was assessed using the DAS28-CRP scale, this involved the counting of tender and swollen joints by an experienced rheumatologist. Patients were excluded if they had chronic heart failure, cancer, chronic kidney disease, HIV, or if they were receiving biological drugs for RA. All the patients presented with mild-to-moderate disease activity, corresponding to a score between 2.6 and 5.1, as assessed by the DAS28-CRP scale, and were on stable baseline therapy without changes in pharmacological treatment for the three months prior to recruitment or during their participation in the study. A group of 12 healthy controls was included to assess their baseline plasma expression of miRNAs as compared with those of the RA patients. All the study participants agreed to participate and signed informed consent prior to the sample.

### 2.2. Plant-Based Diet

A personalized, 14-day isocaloric PBD plan was implemented for each patient, based on their habitual diet, determined by three 24 h dietary recalls (two weekdays and one weekend day). The energy requirements were calculated using the Harris–Benedict formula to estimate the Resting Energy Expenditure (REE), and the Total Energy Expenditure (TEE) was adjusted based on the individual’s physical activity level. The isocaloric plan was non-restrictive in terms of caloric intake, ensuring that the caloric intake met each patient’s energy requirements. The macronutrient distribution was set at 57% carbohydrates, 28% lipids, and 17% proteins, with 80% of the protein sourced from plant-based foods, including legumes (beans, lentils), cereals (corn, oats), seeds, and vegetables (squash, cactus). Animal products were limited to 20% of the total protein intake, sourced from white cheeses (e.g., panela), eggs, and fish.

Processed foods, refined sugars, and saturated fats were excluded from the diet in line with established guidelines to minimize pro-inflammatory components. To ensure the nutritional adequacy of the diet, adherence was monitored using a food consumption diary, based on the National Research Council (NRC) guide (1989). Adherence was evaluated by comparing the actual intake with the prescribed diet, ensuring that caloric intake remained within an acceptable range, defined as 80% (under-consumption) to 120% (over-consumption) of the prescribed energy and macronutrient levels. To control the actual food intake, 24 h dietary recalls were conducted at baseline and during the intervention. These recalls, combined with food diaries, enabled the continuous monitoring of the participants’ dietary compliance and allowed for the reinforcement of dietary recommendations during the study.

To check the accuracy of the dietary records, weekly reviews of the food diaries were conducted with each participant. Additionally, 24 h recalls were applied during the intervention period to compare and verify consistency in the reported intake.

### 2.3. Anthropometric Assessment

Anthropometric measurements were conducted at two time points: at the beginning and at the end of the dietary intervention. All the measurements were taken under fasting conditions and at the same time of day for each patient, following the recommended protocols for bioelectrical impedance analysis (BIA). Bioimpedance was measured using the OMRON HBF-514C (Omron Healthcare Co., Ltd., Kyoto, Japan), which provided data on body weight, Body Mass Index (BMI), percentage of body fat, and visceral fat percentage.

Height was measured using a Seca stadiometer, with the patients standing barefoot and maintaining an upright position. Waist and hip circumferences were measured using a Lufkin body tape measure. The waist-to-hip ratio was calculated by dividing the waist circumference by the hip circumference, following standardized procedures. Each measurement was taken twice to ensure accuracy, and the average value was recorded.

### 2.4. Blood Sample Collection

Peripheral blood samples were collected from RA patients and control subjects in a fasting state, using aseptic techniques through venipuncture both at baseline (before the intervention) and after 14 days of the dietary intervention. A total volume of 4 mL of peripheral venous blood was drawn into a vacutainer tube containing ethylenediaminetetraacetic acid (EDTA) for the separation of whole blood and plasma.

On the same day, an additional blood sample from each patient was sent to the central laboratory of the institute for biochemical analysis. The biochemical profiles measured included glucose, uric acid, triglycerides, total cholesterol, and C-reactive protein (CRP), all of which were analyzed using standardized automated methods. Additionally, the erythrocyte sedimentation rate (ESR) was determined using the Westergren method. All the measurements were performed according to the laboratory’s internal quality control procedures to ensure precision and accuracy.

### 2.5. Selection of miRNAs

The selection of miRNAs was performed through a search of expression data in the Gene Expression Omnibus (GEO) information repository (https://www.ncbi.nlm.nih.gov/geo/, accessed on 23 April 2025) of the National Center for Biotechnology Information (NCBI), performing the search under the concepts of rheumatoid arthritis (RA), differential expression, and microarrays. Files in CEL format with the access number GSE124373 were retrieved, and a differential expression analysis of miRNAs was performed between healthy controls and patients with RA. Subsequently, a search was conducted in PubMed for miRNAs that have been associated with RA, generating a list of all the miRNAs described to date.

The final selection was based on miRNAs that were significantly differentially expressed (|log2FC| ≥ 0.5, *p* < 0.05) and had at least one documented association with RA in the literature.

### 2.6. Prediction of miRNA Target Genes

Target gene prediction for the selected miRNAs was performed using the algorithms in the miRDB (https://mirdb.org/), TargetScan (https://www.targetscan.org/vert_80/, accessed on 23 April 2025), miRWalk (http://mirwalk.umm.uni-heidelberg.de/, accessed on 23 April 2025), and miRmap (https://mirmap.ezlab.org/, accessed on 23 April 2025) databases. These databases are based on identifying the nucleotide pairing between the 3′UTR region of a target mRNA and the 5′ “Seed” region (2–7 nucleotides) of an miRNA. The cutoff criterion for the selection of miRNA targets was that they were present in at least three of the analyzed databases.

### 2.7. Expression Microarray Analysis

miRNA data were obtained in CEL format from the GeneChip miRNA 4.0 microarray (Affymetrix, Santa Clara, CA, USA) with the accession number GSE124373. The files were processed using the Robust Multiarray Analysis (RMA) method in the R-BiocMananger environment. On the other hand, differentially expressed genes (DEGs) between the control group and the case group (RA) were retrieved in CEL format from the GeneChip HGU133plus microarray (Affymetrix) with the accession number GSE206848 and analyzed using the RMA method. This secondary dataset was not used for external validation, but rather to integrate transcriptomic data by identifying shared mRNA targets potentially regulated by the RA-associated miRNAs selected in this study.

In both datasets, genes or miRNAs that presented −log2FC values ≤ −0.5 or ≥+0.5 and *p*-value < 0.05 were considered differentially expressed.

### 2.8. Selection of Differentially Expressed Genes (DEGs)

Candidate genes were selected through a comparative analysis of predicted targets for the selected miRNAs and DEGs from the microarray analysis. Genes shared between the two datasets were represented using a Venn diagram, ensuring that shared genes were associated with RA and were targets of RA-related miRNAs.

### 2.9. Analysis of Signaling Pathways

The set of genes shared between miRNA targets and DEGs in RA was analyzed using the Kyoto Encyclopedia of Genes and Genomes (KEGG) database to identify select signaling pathways involved in RA development. Representative graphs of the signaling pathways in Homo sapiens were created using ShinyGO v0.8 Enrichment analysis + more software (https://bioinformatics.sdstate.edu/go80/, accessed on 23 April 2025). Genes involved in the selected signaling pathways were subsequently retrieved and used to form an interaction network between miRNAs and their respective targets using Cytoscape v3.7.2 software to identify candidate genes that interact in more than one signaling pathway.

### 2.10. miRNA Quantification

For miRNA quantification, 100 μL of plasma was used for miRNA detection, following the protocols of the Zymo Direct-zol RNA Microprep kit (catalog number: R2062, Zymo Research Corp., Irvine, CA, USA) for RNA extraction. Reverse transcription was performed using the TaqMan MicroRNA Reverse Transcription Kit Applied Biosystems, Foster City, CA, USA), followed by the TaqMan MicroRNA assay (Applied Biosystems, Bedford, MA, USA).

The following TaqMan miRNA assays were used: hsa-miR-16-5p (Assay ID: 000391, Catalog number: 4427975), hsa-miR-26a-5p (Assay ID: 000405, Catalog number: 4427975), hsa-miR-125a-5p (Assay ID: 002198, Catalog number: 4427975), hsa-miR-125b-5p (Assay ID: 000449, Catalog number: 4427975), hsa-miR-146a-5p (Assay ID: 000468, Catalog number: 4427975), and hsa-miR-155-5p (Assay ID: 002623, Catalog number: 4427975). Reverse transcription (RT) reactions were performed with custom stem–loop primers (Applied Biosystems) specific to the mature miRNA sequence, which was obtained from miRBase (http://www.miRBase.org, accessed on 3 June 2013). Amplification reactions were performed using the CFX Oppus 96 real-time PCR system (Bio-Rad, Hercules, CA, USA).

The quantitative RT-PCR data were analyzed using the comparative threshold cycle (Ct) method, with cel-miR-39 used as an exogenous control. Relative miRNA expression levels were calculated using the 2^−ΔΔCt^ formula.

### 2.11. Statistical Analysis

The Shapiro–Wilk test was used to assess the normality of the variables. Data are reported as mean ± standard deviation (SD) if normally distributed or as the median and interquartile range (IQR) if not, while categorical variables are presented as frequencies and percentages. Mann–Whitney U was used to test differences between apparently healthy subjects and those with baseline RA, whereas the Wilcoxon test was used to assess changes from baseline to final measurements. The statistical analysis was performed using the Statistical Package for the Social Sciences version 26 (IBM, Armonk, NY, USA) and GraphPad Prism version 8.1 (GraphPad Software; La Jolla, CA, USA), which were utilized for data calculations and visualization. A *p*-value < 0.05 was considered statistically significant.

## 3. Results

After treatment, the RA patients showed reductions in DAS28-CRP-mediated disease activity (*p* < 0.0001) (Table 1), body weight (*p* = 0.014), BMI (*p* = 0.001), glucose levels (*p* = 0.022), CRP levels (*p* = 0.020), total cholesterol (*p* = 0.0034), as well as the number of swollen joints (*p* = 0.005) and tender joints (*p* < 0.0001) (Table 1). It is worth mentioning that all the patients were undergoing stable pharmacological treatment two months prior to and during the 14-day intervention. Additionally, the patients report no discomfort due to the intervention. Additional information about the characterization of the participants is presented in Appendix A.

The microarray analysis showed 91 differentially expressed miRNAs (DEmiRNAs) that met the cutoff criteria (−log2FC ≤ 0.5 or ≥0.5 and a *p*-value < 0.05). From this dataset, 84 miRNAs were found to be downregulated and 7 were upregulated (Figure 1a). Comparative analysis between DEmiRNAs and miRNAs associated with RA in the literature showed 16 miRNAs shared between both datasets, from which the following were selected: hsa-miR-16-5p, hsa-miR-26a-5p, hsa-miR-125a-5p, hsa-miR-125b-5p, hsa-miR-146a-5p, and hsa-miR-155-5p, due to their relevance in the development of RA (Figure 1b). Target gene prediction analysis showed a total of 1676 target genes for the selected miRNAs: miR-16-5p (545 genes), miR-26a-5p (384 genes), miR-125a-5p (468 genes), miR-125b-5p (454 genes), miR-146a-5p (151 genes), and miR-155-5p (245 genes) (Figure 1c).

The HGU133plus 2.0 expression microarray analysis showed a total of 2349 DEGs that met the previously established cutoff criteria (−log2FC ≤ 0.5 or ≥0.5, a *p*-value < 0.05), of which 1206 were found downregulated and 1143 upregulated (Figure 1d). This group of genes was compared with 1677 target genes identified for the six selected miRNAs. Through comparative analysis, 212 genes were identified as being present in both datasets (Figure 1e). The set of shared genes was analyzed to identify signaling pathways involved in RA. Through the ShinyGO v0.8 and KEGG pathway predictive algorithm, 20 enriched signaling pathways in which the DEGs participate were identified (Figure 1f). Subsequently, 39 genes involved in these signaling pathways were recovered, with which an interaction network between miRNAs and their respective target genes was formed (Figure 2).

When comparing the miRNA plasma levels in apparently healthy subjects vs. the baseline subjects in the RA group, it was observed that miR-16, miR-26a-5p, miR-125a-5p,146a-5p, and miR-155 levels were higher in RA patients as compared with the control group, except for miR-125b-5p levels (Figure 3). Interestingly, after 14 days of the PBD, a decrease in the plasma levels of miR-26a, miR-125a, and miR-155 was observed. However, no differences were found in the levels of miR-16, miR-125b, and miR-146a.

## 4. Discussion

This pilot study demonstrates that a short-term, isocaloric PBD intervention may lead to significant improvements in clinical activity and inflammatory biomarkers in patients with rheumatoid arthritis, alongside the modulation of specific circulating microRNAs implicated in the pathogenesis RA. Notably, the plasma levels of miR-26a-5p, miR-125a-5p, and miR-155-5p significantly decreased after 14 days of dietary intervention, paralleling reductions in DAS28-CRP, CRP, glucose, and total cholesterol.

Our findings align with previous studies that have shown that plant-based diets can positively influence RA disease activity and metabolic parameters. The significant reduction in clinical variables such as joint counts and CRP reflects a decrease in RA activity, consistent with findings from other populations. Furthermore, we demonstrated that an isocaloric version of this diet is adaptable to our population and can elicit beneficial effects [16].

In addition, our results showing the elevation of most of the selected miRNAs in the plasma of RA patients corroborate previousstudies that have established these miRNAs as reliable biomarkers for RA [22,23,24]. Ormseth et al. reported that the plasma concentrations of these miRNAs significantly increase in RA patients compared with healthy controls [25]. These miRNAs that changed during our study have previously been positively correlated with disease activity (DAS28-CRP), ESR, and CRP [22,26]. Furthermore, regarding the principal finding of this study—the decrease observed in miR-26a-5p, miR-125a-5p, and miR-155-5p levels in plasma—it is worth mentioning that, in a long-term dietary intervention study, miR-26a was downregulated in groups following a Mediterranean diet enriched with olive oil or nuts compared with that in a group on a low-fat diet [27]. Similarly, in a study of healthy women on a diet enriched with polyunsaturated fatty acids, miR-125a-5p levels decreased after eight weeks [28]. These results suggest that miR-125a-5p expression can also be modified by diets rich in polyunsaturated fats, which are key components of PBDs. Furthermore, miR-155 expression decreased in individuals with metabolic syndrome after following a hypocaloric Mediterranean diet for one week [29]. Also, miR-125a-5p and miR-155 have been implicated in both macrophage inflammation and cholesterol metabolism, particularly the uptake of oxidized LDL (ox-LDL) [30,31,32]. Additionally, miR-26a-5p has been shown to promote proliferation, invasion, and apoptosis resistance in rheumatoid arthritis synovial fibroblasts (RA-FLS) through the PTEN/PI3K/AKT pathway [33]. Elevated miR-26a levels can also exacerbate cartilage destruction by regulating chondrocyte proliferation and apoptosis via the CTGF (Connective Tissue Growth Factor) pathway [34]. However, we cannot discard the possibility that some of the effects were due to an enhanced response to the underlying pharmacological therapy.

Importantly, this study was designed with bioinformatic rigor, leveraging publicly available datasets to identify RA-relevant miRNAs and their validated target genes. The subsequent pathway analysis revealed that the target genes of these miRNAs were involved in key inflammatory cascades, such as the PI3K-Akt, NF-κB, and cytokine–cytokine receptor interactions, providing plausible mechanistic explanations for the observed clinical improvements (Appendix A).

Despite the promising results, this study presents several important limitations. The small sample size and absence of a control group limit the generalizability of the findings and preclude definitive conclusions regarding causality. Another limitation is the absence of a parallel control group receiving a non-intervention diet, which restricts the attribution of the effects solely to the plant-based intervention. Additionally, the short duration of the intervention (14 days) may not adequately capture the long-term effects of dietary modulation on miRNA expression or disease progression. Functional validation of the identified miRNA–target gene interactions was not performed, and the observed changes may have been influenced by external factors such as medication adherence or caloric variability. Although dietary adherence was monitored, variations in individual compliance with the prescribed plant-based diet could have influenced the observed outcomes. Moreover, the predominance of female participants may restrict the applicability of the results to male RA populations. Finally, multiple comparisons were not adjusted in this pilot study, given its exploratory design.

Future research should validate these findings in larger, randomized controlled trials with extended follow-up periods. Integrating functional assays to experimentally confirm miRNA–target interactions, identifying the cellular origins of circulating miRNAs, and stratifying patients based on clinical response could provide deeper mechanistic insights. Ultimately, combining PBDs with established pharmacological therapies may provide a more comprehensive and personalized approach to the management of RA.

## 5. Conclusions

This pilot study suggests that a short-term plant-based diet may beneficially modulate the circulating levels of rheumatoid arthritis-associated microRNAs, with concurrent improvements in clinical and inflammatory markers.

These preliminary findings support further research on the immunomodulatory potential of plant-based diets as complementary strategies in rheumatoid arthritis management, while acknowledging the study’s limitations.

## Figures and Tables

**Figure 1 nutrients-17-02222-f001:**
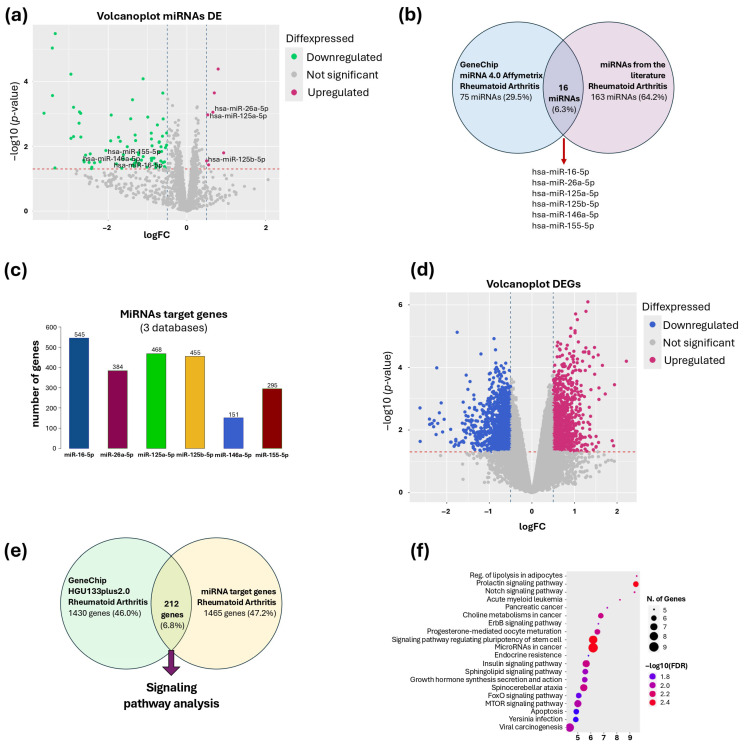
Bioinformatic analysis of selected miRNAs. (**a**) Volcano plot of differentially expressed miRNAs, showing upregulated (hsa-miR-26a, hsa-miR-125a-5p, and hsa-miR-125b-5p) and downregulated (hsa-miR-16-5p, hsa-miR-146a-5p, hsa-miR-155-5p) miRNAs. (**b**) Venn diagram of DEmiRNAs from the microarray and miRNAs reported in the literature. (**c**) Target genes of miRNAs are present in at least three databases. (**d**) Volcano plot of HGU133plus2.0 gene expression microarray. (**e**) Venn diagram of DEGs from the microarray and miRNA target genes. (**f**) Signaling pathways associated with RA.

**Figure 2 nutrients-17-02222-f002:**
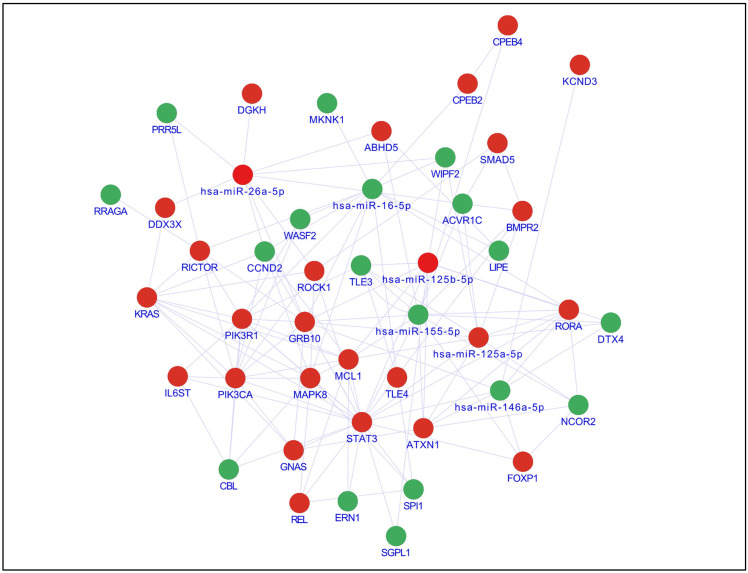
Interaction between miRNAs and their potential target genes; solid lines represent direct interaction. Genes in green are underexpressed, while genes in red are overexpressed. miRNAs are shown in blue.

**Figure 3 nutrients-17-02222-f003:**
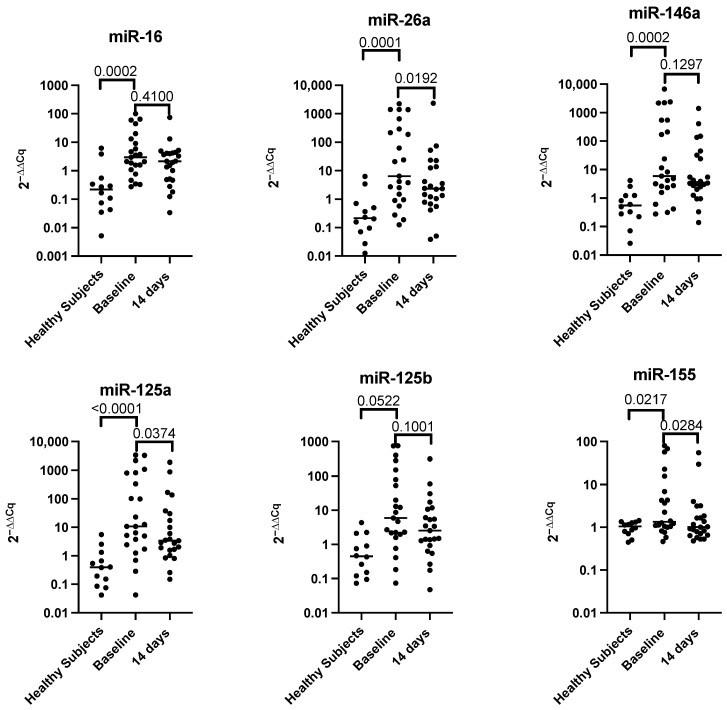
The scatter plots show the median of the relative abundance of miRNAs in 12 apparently healthy subjects and 23 patients at baseline and 14 days after PBD. The miRNA abundance in plasma was measured by RT-qPCR using cel-miR-39 as a reference control, and the 2^−ΔΔCt^ formula was used to estimate relative abundance. Differences between healthy subjects versus baseline were analyzed using the Mann–Whitney test, whereas baseline vs. 14 days were evaluated using the Wilcoxon signed-rank test showing the exact *p*-values using GraphPad Prism, version 8.1.

**Table 1 nutrients-17-02222-t001:** Clinical, anthropometric, laboratory measurements before and after dietary intervention with a plant-based diet.

*N* = 23	Baseline	14 Days	*p*
Female/Male sex	22/1		
Age (years)	56 (51–63)		
Weight (kg)	65.50 (60.75–83.05)	64.70 (59.25–83.65)	**0.014**
Waist-to-hip radio	0.88 (0.84–0.94)	0.89 (0.84–0.92)	0.313
BMI (kg/m^2^)	29.50 (25.80–33.05)	29.20 (25.15–32.75)	**0.001**
% Body fat	44.40 (37.10–48.05)	45.00 (37.25–49.40)	0.736
%Visceral fat	10.00 (7.50–12.00)	10.00 (7.00–12.50)	0.052
Serum glucose (mg/dL)	92.00 (82.50–104.00)	87.00 (80.00–99.00)	**0.022**
Serum uric acid (mg/dL)	4.91 (3.90–5.81)	4.98 (4.19–5.61)	0.543
Total cholesterol (mg/dL)	180.00 (144.00–211.00)	155.00 (141.00–199.00)	**0.0034**
HDL-C (mg/dL)	47.50 (41.65–60.40)	46.50 (41.10–56.80)	0.363
Triglycerides (mg/dL)	134.00 (106.00–174.00)	130.00 (107.50–176.00)	0.243
CRP (mg/L)	5.61 (3.38–8.96)	4.78 (2.35–7.40)	**0.020**
ESR (mm/h)	17.00 (7.50–33.50)	15.00 (8.00–25.00)	0.149
Swollen joints	5.00 (3.00–8.00)	3.00 (1.50–4.50)	**0.005**
Tender joints	7.00 (2.50–8.00)	3.00 (1.00–3.50)	**<0.0001**
DAS28-CRP	4.04 (3.33–4.72)	3.43 (2.92–3.60)	**<0.0001**

Data are presented as median (interquartile range) for continuous variables. The differences between the initial and final values of the evaluated variables were analyzed using the Wilcoxon test. ESR = erythrocyte sedimentation rate. DAS28-CRP = disease activity score for 28 joints based on the C-reactive protein level. Significant *p* value in bold.

## Data Availability

Raw data are available upon reasonable request from the corresponding author.

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
