# Peer review of "A 14-Day Plant-Based Dietary Intervention Modulates the Plasma Levels of Rheumatoid Arthritis-Associated MicroRNAs: A Bioinformatics-Guided Pilot Study"

_nutrients, 2025, doi:10.3390/nu17132222_

Round 1
Reviewer 1 Report
Comments and Suggestions for Authors
Please see attached document.

Author Response
REVIEWER 1
Suggestion: minor revisions
- Please provide evidence for your statement in lines 65-67.
R= Thank you for your suggestion. We have addressed this point by dividing the sentence into two parts. The first part maintains reference to conventional pharmacologic treatments [3], while the second introduces three original randomized controlled trials that support the beneficial effects of dietary interventions as adjunctive strategies in rheumatoid arthritis management (Walrabenstein et al., 2023; Hartmann et al., 2022; Vadell et al., 2020) References 4-6.
- Please provide the following information in tables (you can add these as supplementary tables if it is easier):
- For all 23 patients – what drug intervention they were on (name of drugs), were they smoking (yes/no)? Consuming alcohol (yes/no)? and % change in the 3 parameters of DAS28-CRP, CRP and ESR post treatment ((value post PBD/value pre PDB) *100)
R= We thank you for this detailed suggestion. We have compiled the requested clinical information for all 23 patients, including pharmacological treatment, smoking and alcohol consumption status, as well as percentage changes in DAS28-CRP, CRP, and ESR following the intervention. This information has been added as Supplementary Table 1.
- For all the microRNAs in figure 3 – p value post PBD, biological role/significance, pathways involved and references for the same
R= We thank you for this detailed suggestion. As suggested, we have prepared a table summarizing for each microRNA the post-intervention p-value, reported biological significance in rheumatoid arthritis or inflammation, implicated signaling pathways, and supporting references. This table is now included as Supplementary Table 2.
- Please provide a flowchart of all the methods used in a single figure – to make it easier for your readers to grasp your
We agree with your suggestion and have developed a methodological flowchart summarizing the main steps of the study, from bioinformatic analysis and patient recruitment to miRNA quantification and data integration. This has been included as a Graphical abstract.
Reviewer 2 Report
Comments and Suggestions for Authors
I had the pleasure to review the manuscript ' A14-day plant -based dietary intervention modulates plasma levels of rheumatoid arthritis-associated microRNAs: a bioinformatics-guided pilot study'.
In the abstract line 43 the wortd 'assess' was used twice in a row.
The introduction is well written.
The lines 94-98 in the introduction present the aim of the work.
I would prefer the aim to be written as a separate paragraph- it would be more clear.
It is an interseting study with participation of 23 patients with rheumatoid arthritis and 12 healthy controls. Why the groups are uneven? How the number of participants was calculated? -these questions need to be answered befor publication. There is no information however if the participants received some free plant-based food of they just declared in their food consumption diary that they ate foods of plant origin. My point is how the researches controlled what paticipants really ate? How did they check if their answers in the diaries were true? Was there any validation?
Participants were 51-63 years olds; Mainly women- Which means that they were postmenopausal women, while rheumatoid arthritis (RA) is diagnosed in 16-year-olds and all the older age groups. I mean that the study group was not representative for the population of women with RA. There should be women from different age groups.
The researchers found a significant decrease in body mass , BMI, serum glucose concentration, total cholesterol level in the blood serum, CRP in the blood, the number of swollen and tender joints as well as DAS28 in the study group. The athropometric and blood test results of the control group are missing in Table 1. The circulating microRNAs (miR-26a-5p, miR-125a-5p, miR-155-5p ) significantly decreased after the 14-day plant-based diet in the study group.
Authors indicate a link between dietary patterns and gene regulatory mechanisms. In my opinion the conclusions should be written in 1 or 2 points.
The references were cited correctly.
Author Response
REVIEWER 2
- I had the pleasure to review the manuscript' A14-day plant -based dietary intervention modulates plasma levels of rheumatoid arthritis-associated microRNAs: a bioinformatics-guided pilot study'. In the abstract line 43 the wortd 'assess' was used twice in a row.
R= Thank you for the comment. We corrected the typographical error in the abstract where the word “assess” appeared twice. The revised sentence now reads: “...were conducted to assess the biological relevance of these findings.”
- The introduction is well written. The lines 94-98 in the introduction present the aim of the work. I would prefer the aim to be written as a separate paragraph- it would be more clear.
R=Thank you for the comment. We have revised the introduction and placed the study aim in a separate paragraph, as suggested.
- It is an interseting study with participation of 23 patients with rheumatoid arthritis and 12 healthy controls. Why are the groups uneven? How was the number of participants calculated?
We thank the reviewer for this important and insightful observation.
As indicated in the manuscript title and described in the Methods section, this study was designed and conducted as a pilot, exploratory investigation, primarily aimed to explore and validate, through a bioinformatics-guided approach, the regula-tion of key RA-related miRNAs in response to a 14-day PBDs intervention, and to evaluate their potential relationship with changes in clinical and biochemical parame-ters in patients with rheumatoid arthritis. Given this pilot nature, no formal a priori sample size calculation was performed.
Instead, we relied on recommendations from the methodological literature on minimum sample sizes appropriate for pilot studies. According to Johanson and Brooks (2010), for preliminary psychometric testing and feasibility studies, a sample size of approximately 30 participants per group is commonly considered a cost-effective and methodologically sound threshold [1]. Furthermore, van Iterson and colleagues (2009) provide guidance specific to gene expression profiling, indicating that for moderate statistical power (≥0.6), a sample size of 10–12 biological replicates per group may be sufficient, depending on platform sensitivity and expected effect size [2]. Our study included 23 RA patients and 12 healthy controls, which aligns with these guidelines.
Nonetheless, we fully recognize these limitations which are mentioned in the limitations sections that highlight the exploratory purpose and the need for larger, powered studies to confirm these preliminary findings.
References:
[1] Johanson, G. A., & Brooks, G. P. (2010). Initial Scale Development: Sample Size for Pilot Studies. Educational and Psychological Measurement, 70(3), 394–400. https://doi.org/10.1177/0013164409355692
[2] van Iterson, M., 't Hoen, P., Pedotti, P., et al. (2009). Relative power and sample size analysis on gene expression profiling data. BMC Genomics, 10, 439. https://doi.org/10.1186/1471-2164-10-439
- There is no information however if the participants received some free plant-based food of they just declared in their food consumption diary that they ate foods of plant origin. My point is how the researches controlled what paticipants really ate?
R= Thank you for this question. To control the participants’ actual intake, 24-hour recalls were applied at two points: before the intervention, to estimate the qualitative and quantitative characteristics of their habitual diet and to prescribe an individualized, isocaloric, plant-based diet; and also during the intervention, with the aim of monitoring actual consumption, reinforcing dietary instructions, and confirming the intake reported in the food diary. This strategy made it possible to continuously verify what the participants were truly eating.
- How did they check if their answers in the diaries were true?
R= Thank you for the comment. To verify the accuracy of the responses recorded in the food diaries, weekly reviews of the records were carried out with each participant, complemented by 24-hour recalls during the intervention. This approach made it possible to compare and validate the information entered in the diaries, identify discrepancies or possible omissions, and provide immediate feedback to ensure data quality and adequate adherence to the prescribed diet.
- Was there any validation?
R= Thank you for the comment. The dietary data were monitored through food diaries and 24-hour recalls, reviewed with the support of validated questionnaires routinely applied by our institution’s nutrition department. However, no additional objective validation procedures were performed specifically within this study, which is acknowledged as a limitation.
- Participants were 51-63 years olds; Mainly women- Which means that they were postmenopausal women, while rheumatoid arthritis (RA) is diagnosed in 16-year-olds and all the older age groups. I mean that the study group was not representative for the population of women with RA. There should be women from different age groups.
R= Thank you for the comment. It is correct that the median age of our participants was 56 years (IQR 51–63), which corresponds to predominantly postmenopausal women. While rheumatoid arthritis can indeed be diagnosed as early as 16–18 years of age, its peak incidence occurs in the sixth decade of life (50–59 years), which is consistent with the age of our study population. This pattern has been reported in previous cohorts:
Carbonell-Bobadilla N, Soto-Fajardo C, Amezcua-Guerra LM, Batres-Marroquín AB, Vargas T, Hernández-Diazcouder A, Jiménez-Rojas V, Medina-García AC, Pineda C, Silveira LH. Patients with seronegative rheumatoid arthritis have a different phenotype than seropositive patients: A clinical and ultrasound study. Front Med (Lausanne). 2022 Aug 16;9:978351. doi: 10.3389/fmed.2022.978351. PMID: 36052337; PMCID: PMC9424641.
Myasoedova E, Crowson CS, Kremers HM, Therneau TM, Gabriel SE. Is the incidence of rheumatoid arthritis rising?: results from Olmsted County, Minnesota, 1955-2007. Arthritis Rheum. 2010 Jun;62(6):1576-82. doi: 10.1002/art.27425. PMID: 20191579; PMCID: PMC2929692.
- The researchers found a significant decrease in body mass , BMI, serum glucose concentration, total cholesterol level in the blood serum, CRP in the blood, the number of swollen and tender joints as well as DAS28 in the study group. The athropometric and blood test results of the control group are missing in Table 1. The circulating microRNAs (miR-26a-5p, miR-125a-5p, miR-155-5p ) significantly decreased after the 14-day plant-based diet in the study group. Authors indicate a link between dietary patterns and gene regulatory mechanisms. In my opinion the conclusions should be written in 1 or 2 points.
Thank you for the comment. We have revised the conclusion section to present the main messages in two concise points, as suggested, which improve clarity and focus for the reader.
Round 2
Reviewer 2 Report
Comments and Suggestions for Authors
The revised version is good enough for publication.